# Growth and Biocontrol of *Listeria monocytogenes* in Greek Anthotyros Whey Cheese without or with a Crude Enterocin A-B-P Extract: Interactive Effects of the Native Spoilage Microbiota during Vacuum-Packed Storage at 4 °C

**DOI:** 10.3390/foods11030334

**Published:** 2022-01-25

**Authors:** Nikoletta Sameli, John Samelis

**Affiliations:** Dairy Research Department, Institute of Technology of Agricultural Products, Hellenic Agricultural Organization “DIMITRA”, Katsikas, 45221 Ioannina, Greece; nikol.sameli@gmail.com

**Keywords:** whey cheese, Anthotyros, biopreservation, *Listeria monocytogenes*, enterocin A-B-P, *Leuconostoc mesenteroides*, Carnobacterium

## Abstract

Effective biopreservation measures are needed to control the growth of postprocess *Listeria monocytogenes* contamination in fresh whey cheeses stored under refrigeration. This study assessed growth and biocontrol of inoculated (3 log_10_ CFU/g) *L. monocytogenes* in vacuum-packed, fresh (1-day-old) or ‘aged’ (15-day-old) Anthotyros whey cheeses, without or with 5% of a crude enterocin A-B-P extract (CEntE), during storage at 4 °C. Regardless of CEntE addition, the pathogen increased by an average of 2.0 log_10_ CFU/g in fresh cheeses on day 15. Gram-negative spoilage bacteria also increased by an average of 2.5 log_10_ CFU/g. However, from day 15 to the sell-by date (days 35–40), *L. monocytogenes* growth ceased, and progressively, the populations of the pathogen declined in most cheeses. This was due to an unmonitored, batch-dependent natural acidification by spoilage lactic acid bacteria, predominantly *Leuconostoc mesenteroides*, which reduced the cheese pH to 5.5, and finally to ≤5.0. The pH reductions and associated declines in pathogen viability were greater in the CEntE-treated samples within each batch. *L. monocytogenes* failed to grow in cheeses previously ‘aged’ in retail for 15 days. Overall, high population levels (>7.5 log_10_ CFU/g) of psychrotrophic *Enterobacteriaceae*, particularly *Hafnia alvei*, were associated with an extended growth and increased survival of *L. monocytogenes* during storage.

## 1. Introduction

*Listeria monocytogenes* remains a food-borne pathogen of great concern for the dairy industry [1,2]. The public health concerns associated with the continuing prevalence of *L. monocytogenes* in various retail cheeses have been demonstrated by several listeriosis outbreaks, mostly linked to fresh nonripened and/or surface-ripened soft and semisoft cheeses [3,4,5,6]. In general, challenge testing should be done by following the European Commission (2014) technical guidance document on all types of ready-to-eat (RTE) foods that do not comply with the Regulation 2073/2005 safety criteria categories in relation to *L. monocytogenes* and thus may support its growth during retail distribution and storage [7,8,9,10]. In specific, only RTE cheese products that do not support listerial growth, either a priori based on the pH/a_w_ set value criteria or after research-based challenge testing, can harbor a maximum allowable *L. monocytogenes* level of 100 CFU/g during retail shelf life [11]. Numerous early to recent challenge studies have shown that fresh whey cheeses are the riskiest soft RTE cheese products with regard to *L. monocytogenes* outgrowth because of their high pH (>6.0–6.8) and moisture (>60–80%) content that corresponds with water activity values much above the a_w_ 0.94 threshold [12,13,14,15,16,17]. Therefore, the strict alternative microbiological criterion ‘absence of *L. monocytogenes* in 25 g’, also defined as the ‘zero tolerance’ approach, is required for all RTE whey cheeses, especially when these specialty products are distributed and consumed fresh [11].

Fresh whey cheeses are practically free of native microbiota after manufacture because the indigenous bacteria of the whey, or of the raw milk added to the whey, are inactivated during heating at the high temperatures (>80–95 °C) applied to coagulate the water-soluble milk whey proteins, and no starter cultures are applied after heating [15,18]. Afterward, postprocess handling of the curd results in cross-contamination with a diverse native microbiota, mainly psychrotrophic Gram-negative bacteria, lactic acid bacteria (LAB), staphylococci, spore-forming bacilli, clostridia, yeasts and molds [18,19,20,21]. Therefore, fresh whey cheeses are prone to rapid microbial deterioration, especially at abusive storage temperatures [18], while also being an excellent substrate for the growth of pathogen contaminants. In particular, the psychrotrophic pathogen *L. monocytogenes* may outgrow in the absence or low presence of a protective background microbiota in fresh whey cheeses during aerobic or anaerobic cold storage conditions to cause infection. Indeed, whey cheeses have been linked to listeriosis cases and product recalls, particularly the most popular Ricotta cheese, originally from Italy [1,22,23]. Various antimicrobial methods have been tested or applied commercially to control microbial contamination and growth in Ricotta-type whey cheeses, with emphasis on *L. monocytogenes* [14,22,24,25,26]. Biopreservation using natural antimicrobials or protective, bacteriocin-producing (Bac+) LAB cultures is one of the most attractive and consumer-friendly methods to control the growth of *L. monocytogenes* during retail distribution and storage of Ricotta and other fresh whey cheeses [2,17,27]. In specific, effective antilisterial measures (hurdles) are needed for Myzithra, Anthotyros and Manouri, the most popular traditional Greek whey cheese varieties, because all of them supported a prolific (4.2–6.1 log_10_ CFU/g) temperature-dependent growth of an artificial *L. monocytogenes* postprocess contamination during aerobic storage at 5, 12 and 22 °C [13]. Moreover, *L. monocytogenes* and/or other *Listeria* spp. were found to persist naturally in 12.8%, 1.5% and 2.2% of whey cheese samples from the Greek market surveyed in the years 1990, 1996 and 2010, respectively [28,29].

Samelis et al. [15] conducted the first biopreservation study that applied Nisaplin, a commercial nisin concentrate, to control *L. monocytogenes* postprocess contamination in a traditional Greek whey cheese stored at 4 °C in vacuum packages (VPs) for up to 45 days. Nisin was added either to the Anthotyros whey (100 or 500 IU/g) before heating or to the fresh cheese (500 IU/g) before vacuum packaging. In the control cheeses without nisin, *L. monocytogenes* strain Scott A exceeded 7 log_10_ CFU/g after only 10 days of VP storage at 4 °C. All nisin treatments had an immediate lethal effect (0.7–2.2 log reduction) after inoculation, but weak antilisterial effects during storage, except for the treatment with 500 IU/g added to the whey which suppressed pathogen growth below the inoculation level of 4 log_10_ CFU/g throughout storage [15]. An additional technological concern was that Nisaplin reversed the natural spoilage microbiota of Anthotyros from Gram-positive LAB in the control cheeses without nisin to Gram-negative bacteria; this reversal was more pronounced in the most effective Nisaplin treatments [15]. Later efforts to replace Nisaplin with viable cells of the wild, nisin A-producing (NisA+) *Lactococcus lactis* strain genotype M78/M104 able to produce sufficient amounts of nisin in situ in traditional Greek cheese milk fermentations [30,31] were of limited success in fresh Anthotyros cheese. The primary reason was the inability of the NisA+ *L. lactis* strain to grow competitively and produce nisin at 4 °C. Conversely, at 12 °C, the NisA+ strain along with adventitious LAB reduced the pH and the sensory quality of the fresh Anthotyros; those defects counteracted the benefits associated with an enhanced inhibition of *L. monocytogenes* growth at abusive storage conditions [32].

Overall, a reduced in situ bacteriocin (nisin) production due to a lack of growth of the Bac+ (Nis+) culture at refrigeration temperatures remains one of the main concerns regarding the failure of typical mesophilic dairy-protective strains to control the growth of *L. monocytogenes* in cold-stored soft cheeses [2,33,34]. Alternate biopreservation methods for fresh whey cheeses are the application of psychrotrophic Bac+ strains of nonaciduric LAB, such as *Carnobacterium* spp. [25,26], or the addition of heat-stable (class I or II) LAB bacteriocin concentrates, other than nisin, active against *L. monocytogenes* and spoilage non-LAB bacteria, such as enterocins. An increasing number of promising applications of enterocin-producing (Ent+) *Enterococcus* spp. strains or enterocin preparations in dairy foods are discussed in recent comprehensive reviews [27,35,36], including studies on fresh whey cheeses [17]. In particular, the addition of purified, semipurified or crude enterocin extracts may be a more effective and permissible biopreservation method than the direct addition of viable Ent+ antilisterial cultures in fresh soft (whey) cheeses stored under refrigeration because *Enterococcus* spp. are not psychrotrophic bacteria and the genus does not have a GRAS status [36].

In a previous study, we assessed the effects of a crude enterocin A-B-P extract on the evolution and species composition of the specific spoilage microbiota prevailing in fresh vacuum-packaged Anthotyros whey cheeses stored at 4 °C [37]. This research is a follow-up challenge study to evaluate growth interactions of *L. monocytogenes* with the specific spoilage bacterial species during refrigerated (4 °C) storage of fresh vacuum-packaged Anthotyros whey cheeses without or with enterocin A-B-P supplementation. It has been noted that beyond the in situ production or external addition of bacteriocins, the native microbial community as a whole can also be important in preventing the growth of *L. monocytogenes* in fresh soft cheeses [2]. However, neither sole competitive effects of specific cheese spoilage bacteria against *L. monocytogenes* nor potentially enhanced interactive antilisterial effects of the native microbiota associated with the use of enterocin(s) have so far been evaluated in fresh (Greek) whey cheeses; therefore, this was the dual aim of the present study.

## 2. Materials and Methods

### 2.1. Preparation of the Crude Enterocin A-B-P Extract

The preparation of the crude enterocin A-B-P extract (CEntE) and most methods followed in this study were according to the procedures described by Sameli et al. [37]. Additional experimental methods, such as the whey cheese inoculation with *L. monocytogenes* and additional analyses relating to the interactive growth of the pathogen with the whey cheese spoilage microbiota, are described in later paragraphs.

Briefly, 150 mL of filter-sterilized CEntE was prepared by combining equal (50 mL) volumes of the cell-free supernatant (CFS) of fresh (24 h at 30 °C) MRS broth (Neogen Culture Media, formerly Lab M, Heywood, UK) cultures of three Ent+ *Enterococcus faecium* strain biotypes originally isolated from Greek Graviera cheese: the EntA+ *E. faecium* KE64 (GenBank Acc. No MW644963), the multiple Ent+ (*entA-entB-entP*) *E. faecium* KE82 (MW644969) and the multiple Ent+ (*entA-entB-entP*) *E. faecium* KE118 (not yet deposited in GenBank) [38]. The same CEntE (150 mL) used by Sameli et al. [37] was also used in the challenge experiments of this follow-up study for direct data comparison. The CEntE containing enterocin A (secreted by all three strains) plus enterocins B and P (secreted by KE82 and KE118) was prepared without preceding pH adjustment and stored at –30 °C until use [37]. The Ent+ antilisterial activity, the enterocin titer (400 AU/mL) and the stability of the enterocin titer of the thawed CEntE after every use [37] were determined by the well diffusion assay procedures previously described by Vandera et al. [39].

### 2.2. Commercial Anthotyros Whey Cheese Samples

Eight retail VP samples (500 g each) of fresh Anthotyros cheese, representing four batches previously coded A, B, C and D, were obtained from two semi-industrial dairy plants located in Epirus, Greece [37]. In specific, fresh cheese batches A and B were products of the traditional dairy Pappas Bros. (Skarfi E.P.E., Filippiada, Greece), our collaborator SME in the BIO TRUST project, while batches C and D were fresh retail Anthotyros cheese products of another traditional dairy located near Ioannina.

For the purposes of this challenge study, three additional VP Anthotyros whey cheese batches, representing three independent cheese productions in the second dairy plant, were purchased from the Ioannina market while they had already been stored in retail for 15 days after processing, based on their labeling. These nonfresh (15-day-old) Anthotyros batches, coded X-15A, X-15B and X-15C, were used in the experiments for specific reasons relating to their age and explained in later paragraphs.

### 2.3. Cheese Inoculation, Enterocin Addition and Storage

Within one hour after transportation to our microbiology laboratory in insulated ice boxes, all fresh whey cheese samples (batches A to D) were removed from their original VP films under a laminar flow hood. Then, the 500 g soft cheese masses of the two retail VP samples from each batch were combined aseptically for experimental reasons addressed by Sameli et al. [37]. Next, 50 g portions of soft cheese were transferred by weighing, with the aid of presterilized stainless steel spatulas, into new clean vacuum bags of small size suitable for food storage (Cryovac BK3550 bag; Food Care, Sealed Air Corporation, Milano, Italy). Afterward, all bag samples were artificially contaminated with *L. monocytogenes* strain No. 10 by adding 1 mL of inoculum in each bag, diluted to yield approximately 3 log_10_ of listerial cells/g of cheese. The inoculated cells were from a fresh (24 h; 30 °C) 10 mL culture of *L. monocytogenes* strain No. 10 in brain heart infusion broth (BHI; Neogen), decimally diluted in sterile quarter-strength Ringer solution (Neogen, Lab M) to obtain the cell density for inoculation. *L. monocytogenes* No. 10 (serotype 4ab) is a highly suitable and convenient avirulent target strain, similar to virulent *L. monocytogenes* strains. According to a previous Galotyri PDO cheese challenge study by Sameli et al. [40], strain No. 10 was used as a single target inoculum to exclude potential strain variation effects of *L. monocytogenes* from the Anthotyros whey cheese trials to facilitate the assessment of the CEntE antilisterial effects. For this task, half of the *Listeria*-inoculated, fresh whey cheese bags were vacuum-packaged directly (vacuum level: minus 1 bar; 99.9%) using a MiniPack-Torre, model MVS45L vacuum sealing machine (Dalmine BG, Italy). The remaining half of the bags were vacuum-packaged, as above, following the addition of 5% *(v/w)* CEntE; i.e., 2.5 mL of thawed CEntE was added last in each bag with 50 g of cheese before sealing [37]. All *Listeria*-inoculated cheese samples, with or without CEntE addition, were massaged by hand for 30 s from outside the bag to evenly distribute the inocula and the active enterocin molecules before vacuum packaging. All cheese samples were stored in a refrigerated incubator (Velp Scientifica FOC 225I, Milano, Italy) at 4.0 ± 0.1 °C and analyzed on days 0, 8, 15, 30 and 40 of storage, according to the sampling protocol and the retail shelf life duration of both Anthotyros brand products reported by Sameli et al. [37]

The three ‘aged’ whey cheese batches, X-15A, X-15B and X-15C, were removed from their original bags, distributed in 50 g portions in new bags, inoculated with *L. monocytogenes* No. 10 and vacuum-packaged without previous addition of 5% CEntE by the same procedures described above. This supplementary challenge experiment was conducted to evaluate whether the growth potential and/or survival pattern of *L. monocytogenes* might change in the case cross-contamination of commercial Anthotyros with the pathogen occurs in an ‘aged’ (15-day-old) naturally acidified whey cheese after opening in retail or at home, compared to a fresh (24-hour-old) whey cheese product after manufacture. The VP samples of the ‘aged’ batches were also stored at 4 °C, as above, and analyzed microbiologically and for pH on days 0, 8 and 20 of additional refrigerated storage, which corresponded to a total retail VP storage for 15, 23 and 35 days, respectively, under refrigeration.

### 2.4. Cheese Analyses

For microbial quantification, each VP sample was opened aseptically near a Bunsen burner and 10 g of whey cheese was homogenized with 90 mL of sterilized quarter-strength Ringer solution in stomacher bags (Lab Blender, Seward, London, UK) for 60 s. The homogenates were decimally diluted with Ringer, and duplicate 0.1 mL samples of the appropriate dilutions were spread on total and selective enumeration agar media, all purchased from Neogen Culture Media. The populations of *L. monocytogenes* strain No. 10 were counted on Palcam agar incubated at 30 °C for 48 h; the plates were rechecked at 72 to 96 h of incubation for detection of slow recovery of enterocin-stressed and/or acid-stressed colonies of strain No. 10 in the whey cheese samples. Total mesophilic whey cheese spoilage bacteria were enumerated on Milk Plate Count Agar (MPCA) incubated at 37 °C for 48–72 h. Total psychrotrophic spoilage bacteria were enumerated on Tryptone Soya Agar with 0.6% yeast extract (TSAYE) incubated at 12 °C for 5–7 days. Selective enumerations of the dominant LAB and Gram-negative bacteria colonies were performed on high-dilution MPCA/37 °C and TSAYE/12 °C plates for most samples, as detailed by Sameli et al. [37]. Total aciduric LAB were enumerated on MRS agar incubated at 30 °C for 72 h. *Pseudomonas* and related Gram-negative bacteria were enumerated on cetrimide–fucidin–cephaloridine (CFC) agar, incubated at 25 °C for 48 h [37].

In addition, for the purposes of this study, total mesophilic cheese spoilage bacteria prevailing in the three ‘aged’ whey cheese batches, X-15A, X-15B and X-15C, discriminated in LAB and Gram-negative bacteria colonies as above, were counted on M17 agar plates, incubated at 22 °C for 72 h. Total enterobacteria were enumerated on double-layered Violet Red Bile Glucose (VRBG) agar plates incubated at 37 °C for 24–48 h; enterococci, on Kanamycin Aesculin Azide (KAA) agar incubated at 37 °C for 48 h; total staphylococci, on Baird-Parker agar base with egg yolk tellurite incubated at 37 °C for 48 h; and yeasts, on Rose Bengal Chloramphenicol (RBC) agar incubated at 25 °C for 5 days.

The pH was measured with a digital pH meter (Jenway 3510, Dunmow, Essex, UK) by immersing the electrode in the soft cheese mass after the microbiological sampling.

### 2.5. Isolation and Characterization of the Dominant Whey Cheese Spoilage Microbiota

The most prevalent bacterial species during storage, with emphasis on the end of storage when specific spoilage species had been established in each cheese batch, were isolated and identified. Briefly, the terminal spoilage association of all fresh whey cheese samples after 40 days of VP storage at 4 °C was characterized by purification and biochemical identification of a total of 120 LAB and 96 Gram-negative bacteria isolates from the highest dilution MPCA/37 °C, MRS/30 °C and TSAYE/12 °C plates, i.e., 40 LAB isolates from each agar medium, 30 isolates from each whey cheese batch A to D and 15 isolates from the CN or CEntE sample of each batch [37]. The species identification of 17 LAB isolates selected to represent the most prevalent spoilage LAB biotypes was assured by 16S rRNA gene sequencing; their identities are provided by Sameli et al. [37].

An additional 24 LAB and 12 Gram-negative bacterial isolates from the most diversified whey cheese batches C and D were collected on day 15 of VP storage at 4 °C and characterized to check whether they were similar to, or differed from, the isolates of the specific spoilage LAB or Gram-negative species prevailing in the same batches on day 40. Moreover, 20 LAB and 6 Gram-negative bacterial isolates were recovered from the ‘aged’ X-15A whey cheese batch, as a random representative retail product of the second plant, to check their species similarity to the respective isolates prevailing in the spoiling fresh whey cheese batches C and D on day 15.

Following transfer of their colonies for growth in glass tubes with 10 mL MRS or BHI broth, each of the additional 44 LAB and 18 Gram-negative bacteria isolates was checked for purity by streaking on MRS agar (pH 5.7 ± 0.1) or BHI agar (pH 7.4 ± 0.2) incubated at 30 °C for 72 h or 25 °C for 48 h, respectively; stored in MRS or BHI broth with 20% (*w*/*v*) glycerol at −30 °C; and later characterized, according to the procedures described by Sameli et al. [37]. Briefly, the 44 LAB isolates were first confirmed for their Gram-positive and catalase-negative reactions and then were tested for cell morphology, growth at 37 °C and 45 °C, gas (CO_2_) production from glucose, ammonia (NH_3_) production from arginine, growth in 4% and 6.5% NaCl, slime formation from sucrose and fermentation of 13 differentiating sugars. All Gram-negative bacteria isolates were confirmed for their Gram reaction, separated into oxidase-positive and oxidase-negative isolates and then identified at the genus or species level using the API 20E kit and the corresponding biochemical identification code manual (BioMerieux, Marcy l’ Etoile, Lyon, France), according to the manufacturer’s instructions.

### 2.6. Statistical Analysis

The microbiological data of the fresh whey cheese batches (*n* = 4; main challenge experiment) and of the ‘aged’ whey cheese batches (*n* = 3; supplementary challenge experiment) were converted to log_10_ CFU/g and along with the data for pH were subjected to a one-way analysis of variance using the software Statgraphics Plus for Windows v. 5.2 (Manugistics, Inc., Rockville, MD, USA). The means were separated by the least significant difference procedure at the 95% confidence level (*p* < 0.05) for determining the significance of differences with time of whey cheese storage and between the CN and Ent-treated fresh cheese samples on each sampling interval of the main experiment.

## 3. Results

### 3.1. Behavior of L. monocytogenes in Fresh Anthotyros Whey Cheeses: Potential pH-Dependent Antilisterial Effects of the CEntE during VP Storage at 4 °C

Mean populations of *L. monocytogenes* strain No. 10 increased (*p* < 0.05) by an average of 1.21 log_10_ CFU/g and 1.33 log_10_ CFU/g in the fresh untreated (CN) and the Ent-treated Anthotyros whey cheese samples, respectively, from day 0 to 8 of VP storage at 4 °C (Table 1). Neither the prolific mean growth of the native LAB (1.8–2.4 log_10_ CFU/g increase) and Gram-negative bacteria (1.1–1.8 log_10_ CFU/g increase) in all fresh cheese samples nor the presence of 0.5% CEntE prevented pathogen growth during the first week at 4 °C. Contrary to our experimental anticipation, this early growth of *L. monocytogenes* was not even retarded in the Ent-treated samples compared to the control (CN) samples (Table 1). Instead, the high mean initial pH 6.8 of fresh Anthotyros supported the growth of *L. monocytogenes* (approximately 50-fold increase) while the mean population levels of an actively growing antagonistic LAB biota were still below the 7 log_10_ CFU/g threshold. Accordingly, the initial, almost neutral, pH values underwent minor changes (*p* > 0.05) in most fresh whey cheese samples from day 0 to 8 of storage (Table 1).

*Listeria monocytogenes* continued growth at reduced rates in both treatments from day 8 to 15. This resulted in further population increases of the pathogen, which on average were 0.65 log_10_ CFU/g and 0.73 log_10_ CFU/g in the untreated and the Ent-treated whey cheese samples, respectively (Table 1). On day 15, the mean pH was reduced (*p* < 0.05) to pH 6.2 in the CN samples, and even more (pH 6.0) in the Ent-treated samples. The above pH reductions were due to an unmonitored natural acidification by native LAB which prevailed on days 15, 30 and 40 in all cheese samples and quite faster in the Ent-treated samples on day 15. Gram-negative bacteria also increased well above the 7 log_10_ CFU/g level in most whey cheese samples during storage. However, the prevalence of the antagonistic LAB biota was always higher than the prevalence of Gram-negative bacteria on days 15, 30 and 40 (Table 1). Accordingly, from day 15 to 30, growth of *L. monocytogenes* ceased in most cheese samples, and the pathogen populations generally started to decline. Afterward, from day 30 to 40, except for one sample to be discussed later, the populations of *L. monocytogenes* either remained constant or continued declining while the pH was still decreasing in most naturally acidified whey cheese samples. Overall, pH decreases and consequently the declines of *L. monocytogenes* were greater (*p* < 0.05) in the Ent-treated than in the CN cheese samples on the late days 30 and 40 of storage. Thus, the natural whey cheese acidification and the associated antagonistic effects of LAB against *L. monocytogenes* were somehow stimulated in the presence of the CEntE after the first two weeks of VP storage at 4 °C (Table 1).

At this point, certain major batch-dependent or plant-dependent differences in the growth/survival pattern of *L. monocytogenes* (Figure 1A) should be examined in association with the pH reduction pattern in each whey cheese batch (Figure 1B). First, the significant early growth of the pathogen in all batches supported by the minor changes of their high pH from day 0 to 8 is clearly shown in Figure 1. However, Figure 1A further shows that this early listerial growth was significantly more restricted in batch D, particularly in the CN samples, while it was more pronounced in batch C, particularly in the Ent-treated samples. Notably, batch D and C samples correspondingly had the lowest (pH 6.5–6.6) and the highest (pH 7.1–7.2) pH values on day 8 (Figure 1B). In particular, batch C was the only whey cheese product whose initial pH was increased slightly above 7.0 after the first week of storage (Figure 1B). In fact, although batches C and D were fresh Anthotyros cheese products of the second dairy plant manufactured with only a three-day difference in the plant’s processing line, the initiation of their pH-dependent bacterial spoilage and the support of *L. monocytogenes* growth immediately after processing varied greatly.

Consistent with the above findings, growth continuation of *L. monocytogenes* during the second week (day 8 to 15) of storage was not always significant (Table 1), because the individual pathogen increases differed greatly from batch to batch (Figure 1A). In specific, the growth levels of *L. monocytogenes* became higher by 1–2 log_10_ CFU/g in batches A and C compared to B and D (Figure 1A). In relation, the Ent-treated samples of batch C (pH 6.53) and the CN samples of batch A (pH 6.38) had the highest pH values on day 15 (Figure 1B). Major batch-dependent and/or treatment-dependent fluctuations in *L. monocytogenes* populations continued to occur from day 15 to 30, when, however, the growth of the pathogen ceased and its viable populations started to decline in most batches or treatments (Figure 1A). The most prominent exception was the continuation of major growth of *L. monocytogenes* toward its highest population level of 6.55 log_10_ CFU/g in the untreated (CN) samples of batch C (pH 5.50) on day 30 (Figure 1A). Eventually, with the prominent exception of the CN samples of batch B which supported listerial growth till the end of storage, the viability of *L. monocytogenes* was stabilized (batch D/CN) or underwent further minor (A/CN; B/Ent; C/Ent) or major (A/Ent; C/CN; D/Ent) declines from day 30 to 40, which also varied greatly between batches and treatments (Figure 1A).

In summary, the growth of *L. monocytogenes* was not fully controlled in any of the fresh whey cheese products studied. Indeed, from its lowest population after inoculation (day 0) to its highest batch-dependent population on day 15, or 30 or 40, the total net growth of *L. monocytogenes* in the CN and Ent-treated samples of batches A, B, C and D was 2.62, 2.26, 3.14 and 0.89 log_10_ CFU/g and 2.05, 1.42, 3.23 and 1.53 log_10_ CFU/g, respectively. Eventually, the whey cheese samples formed two distinct clusters on the basis of the terminal *L. monocytogenes* survival on day 40: the first ‘high-risk’ cluster with final pathogen populations ≥5.5 log_10_ CFU/g included batch C and the untreated (CN) batches A and B, whereas the second ‘low-risk’ cluster with final pathogen populations ≤4.5 log_10_ CFU/g included batch D and the Ent-treated batches A and B (Figure 1A). Batch C was the most supportive whereas batch D was the least supportive for *L. monocytogenes* growth, irrespective of CEntE and despite both batches being produced in the same plant. Increased antilisterial effects that were likely to be associated with the addition of 0.5% CEntE occurred in the fresh Anthotyros products of the Pappas dairy only, i.e., in batch A and mainly in batch B (Figure 1A). Overall, significant reductions of *L. monocytogenes* associated with the CEntE occurred after two weeks at 4 °C, provided the cheese pH declined below 5.5. Therefore, on late storage days 30 and 40, the Ent-treated samples of all whey cheese batches clustered at a more acidic pH range than their untreated (CN) counterpart samples (Figure 1B). This finding supports that the growth of native LAB and thereby the natural whey cheese acidification were somehow stimulated in the presence of the CEntE. Stimulation of LAB growth and acidification in turn enhanced the in situ antilisterial activity of the enterocin A, B and P molecules contained in the CEntE after the whey cheese matrix was acidified to pH values from 5.2 down to 4.5 (Figure 1B).

### 3.2. Effects of the Native Spoilage Microbiota on the Growth/Survival Pattern of L. monocytogenes in Fresh VP Anthotyros Whey Cheeses during Storage at 4 °C

Apparently, the aforementioned batch-dependent and plant-dependent differences in the rate and extent of natural acidification, *L. monocytogenes* growth and efficacy of the CEntE were affected by the numbers and types of spoilage bacteria prevailing in each whey cheese batch, without or with CEntE, during VP storage at 4 °C. Therefore, the evolution of specific spoilage species in batches A to D, previously studied in detail by Sameli et al. [37], was reconsidered herein relative to the growth/survival pattern of *L. monocytogenes*. For this purpose, additionally to the total microbial growth data in Table 1, the selective growth on MRS and CFC agar plates of the two main spoilage bacterial groups, LAB and *Pseudomonas*-like bacteria, in each whey cheese batch during VP storage at 4 °C is presented in Figure 2A and Figure 2B, respectively. In addition, an overview of the dominant spoilage LAB and Gram-negative bacteria species in each batch by the end of storage is shown (Table 2). It should be clarified that the graphical data in Figure 2 were previously provided in the supplementary Tables S4 and S5 by Sameli et al. [37].

Starting from the kinetic data, Figure 2 clearly shows the exponential cogrowth of LAB and *Pseudomonas*-like bacteria in all fresh Anthotyros batches during mainly the first (day 8) and the second (day 15) weeks of storage. Additionally, the growth cessation of *Pseudomonas* and related Gram-negative bacteria after day 15 (Figure 2B) while the growth of LAB continued until day 30 (Figure 2A) is evident, in agreement with the corresponding total mean counts in Table 1. However, by examining growth kinetics more carefully in each batch, it can be noted that batch D samples, which supported the least early growth of *L. monocytogenes* (Figure 1A) and had the lowest pH on day 8 (Figure 1B), also had the highest levels of natural aciduric LAB contamination at the beginning of storage (day 0) and the fastest and greatest LAB growth on MRS from day 0 to 15 (Figure 2A). Conversely, all other fresh whey cheese batches had an approximate 2 log_10_ CFU/g lower postprocess LAB contamination on day 0 (Figure 2A), while batches A and B produced in the Pappas dairy plant had an approximate 2 log_10_ CFU/g higher postprocess contamination with *Pseudomonas*-like bacteria (Figure 2B). Hence, after the first week (day 8), the levels of exponentially growing *Pseudomonas*-like bacteria in batches A, B and C ranged from a maximum of 6.5 log_10_ CFU/g to a minimum of 5.5 log_10_ CFU/g (Figure 2B), while the corresponding levels of LAB were lower by approximately 1.5 log_10_ CFU/g, irrespective of CEntE treatment (Figure 2A). After the second week (day 15), growth of *Pseudomonas*-like bacteria in batches A to C increased close to, or above, the 8 log_10_ CFU/g level (Figure 2B), while the growth levels of aciduric LAB on MRS were still 1–1.5 log_10_ CFU/g units lower, particularly in the untreated (CN) samples (Figure 2A). Batch D continued to be an exception because it had 10-fold higher populations of LAB than *Pseudomonas*-like bacteria on day 15 (Figure 2). As a result, growth of *Pseudomonas*-like bacteria in batch D was suppressed one week earlier than in batches A, B and C (Figure 2B) due to the faster growth increases of competitive LAB on days 8 (>7 log_10_ CFU/g) and 15 (> 8 log_10_ CFU/g) of storage (Figure 2A). Regarding the most prominent interactions of the native spoilage microbiota with *L. monocytogenes*, the high (> 4 log_10_ CFU/g) postprocess contamination of batch D with LAB and their early growth above the 7 log_10_ CFU/g threshold (Figure 2A) retarded pathogen growth within the first two weeks of VP storage at 4 °C (Figure 1A). In contrast, the high (>4 log_10_ CFU/g) postprocess contamination and/or the high (>6–8 log_10_ CFU/g) growth of *Pseudomonas* and related Gram-negative spoilage bacteria in batches A to C (Figure 2B) allowed or enhanced the growth of *L. monocytogenes* during early (day 0 to 15) and, in few samples, during late (day 15 to 40) periods of storage (Figure 1A). The fact that the untreated (CN) samples of batches C (mainly) and B continued to support the growth of *L. monocytogenes* after days 15 and 30, respectively, appeared to correlate with the fact that the above two CN batch treatments had the lowest LAB (MRS) counts on day 30 (Figure 2A) while simultaneously having *Pseudomonas* (CFC) counts at the highest (8 log_10_ CFU/g) level (Figure 2B).

Eventually, the whey cheese samples formed two distinct clusters on the basis of the final levels of *Pseudomonas*-like bacteria surviving on days 30 and 40 (Figure 2B), which coincided with the respective two clusters formed by *L. monocytogenes* survivors on day 40 (Figure 1A). The first ‘high-risk’ cluster with final pathogen populations ≥ 5.5 log_10_ CFU/g that included batch C and the untreated (CN) batches A and B (Figure 1A) had final CFC counts ranging from 6.4 to 8.0 log_10_ CFU/g (Figure 2B), whereas the second ‘low-risk’ cluster with final pathogen populations ≤4.5 log_10_ CFU/g that included batch D and the Ent-treated batches A and B (Figure 1A) had final CFC counts ranging from 3.0 to 5.7 log_10_ CFU/g (Figure 2B). Thus, the increased low pH-dependent antilisterial effects correlated strongly with reduced levels of Gram-negative spoilage bacteria in all Ent-treated batches except for batch C, and in batch D irrespective of CEntE addition. By somehow stimulating LAB growth in batches A and B from day 15 to 30 (Figure 2A), the CEntE enhanced natural acidification (Figure 1B) and the declines of *L. monocytogenes* (Figure 1A) and Gram-negative spoilage bacteria (Figure 2B) by late storage. The high final CFC counts in the CN and mainly the Ent-treated samples of batch C (Figure 2B) coincided with the fact that this particular cheese product was the most supportive for *L. monocytogenes* growth (Figure 1A).

The overviews of the spoilage LAB and Gram-negative bacteria species in batches A to D after 30 to 40 days of storage at 4 °C (Table 2), previously detailed by Sameli et al. [37], provide further potential associations with the growth variability of *L. monocytogenes* noted between batches and treatments (Figure 1A). For instance, it is worth noting that (i) indigenous nonaciduric *Carnobacterium maltaromaticum* isolates were recovered from the high-pH batches B and C only (Table 2), both supporting growth of *L. monocytogenes* on late storage days (Figure 1A); (ii) the high terminal levels of Gram-negative spoilage bacteria in batch C (Figure 2B), which was at the highest risk with regard to the outgrowth of *L. monocytogenes* (Figure 1A), consisted mainly of *Hafnia alvei* (Table 2); (iii) the high levels of Gram-negative bacteria in the CN samples of batches A and B that were less supportive for *L. monocytogenes* growth than batch C comprised mainly oxidase-negative fermentative enterobacteria isolates of *Rahnella aquatilis* and *Serratia liquefaciens*, whereas oxidase-positive, nonfermentative *Pseudomonas* spp. isolates were more frequent in the corresponding Ent-treated samples of batches A and B (Table 2), in which the progressive declines of *L. monocytogenes* during storage were much greater (Figure 1A); and (iv) the ‘lowest risk’ batch D contained members of the nonfermentative genera *Aeromonas* and *Flavibacterium* along with another *H. alvei* biotype recovered at 3 to 4 log_10_ CFU/g lower levels than the dominant *H. alvei* biotype in batch C (Table 2).

Overall, regardless of supplementation with the CEntE, the terminal spoilage community of all fresh Anthotyros whey cheese batches was dominated by psychrotrophic, non-slime-producing biotypes of *Leuconostoc mesenteroides* (Table 2). In addition, one typical slime-producing biotype of *Lc. mesenteroides* was recovered mainly from the batch A and B products of the Pappas dairy plant (Table 2). Otherwise, neither did the numerical isolate distribution of the four distinct *Lc. mesenteroides* biotypes in the CN and Ent-treated samples vary greatly nor were other psychrotrophic LAB species exclusively isolated from the Ent-treated samples of batches A and B or from the ‘low-risk’ batch D samples (Table 2) to suggest an increased species-dependent or biotype-dependent antilisterial activity in the above products. However, to our surprise, batch D was the only whey cheese product that contained *Streptococcus thermophilus* after 40 days of VP storage at 4 °C (Table 2). Because proliferate of this thermophilic dairy starter species during fresh Anthotyros whey cheese storage at refrigeration temperatures was impossible, the origin of those dormant *S. thermophilus* isolates as a ‘natural LAB contaminant’ in batch D was assessed and is discussed in Section 3.3 below.

### 3.3. Effects of Anthotyros Aging on the Behavior of L. monocytogenes during VP Storage at 4 °C

Altogether, the results in Figure 1 and Figure 2 reveal that the high growth potential of *L. monocytogenes* in fresh VP Anthotyros products stored at 4 °C was suppressed after the cheese pH declined to pH 5.5, or below, due to the progressive LAB growth causing natural acidification. In most trials, the cold storage time breakpoint for growth cessation of *L. monocytogenes* was day 15. As it was illustrated, earlier growth cessation times were noted in batch D due to the superior early growth of LAB, whereas later growth cessation times were noted in CN samples of batches C (mainly) and B due to the high prevalence of Gram-negative bacteria associated with delayed cheese pH declines. Therefore, it appears that ‘cold aging’ of fresh Anthotyros cheese products per se might prevent the growth of an accidental natural *L. monocytogenes* contamination during retail storage, irrespective of treatment with the CEntE or other preservatives. To validate this, three individual ‘aged’ (15-day-old) retail batches of VP Anthotyros were inoculated with *L. monocytogenes* strain No. 10, without CEntE addition, and analyzed comparatively to the diverse fresh whey cheese batches C to D of the same dairy plant; the results are summarized in Table 3.

As it was anticipated, *L. monocytogenes* strain No. 10 failed to grow (<0.5 log_10_ CFU/g increase) but survived without death in all ‘aged’ whey cheese batches for an additional 20 days, i.e., 35 days of total storage (Table 3). On day 0 of additional storage, the reduced mean pH 6.1 of the three ‘aged’ whey cheese samples reflected the prevalence of a native competitive LAB biota already grown well above 7.5 log_10_ CFU/g, thus preventing the growth of *L. monocytogenes* after day 15 of total storage (Table 3). In agreement with our previous findings [37], the cheeses aged for 15 days under the retail storage conditions contained high levels of total Gram-negative bacteria (7.6 to 7.8 log_10_ CFU/g) and *Pseudomonas*-like bacteria (7.3 to 7.8 log_10_ CFU/g) grown on M17/22 °C and CFC/25 °C, respectively. Another finding in agreement was that the mean levels of total aciduric LAB on MRS/30 °C and coliform bacteria on VRB/37 °C were lower than the levels of total cheese spoilage bacteria on M17/22 °C. Overall, psychrotrophic Gram-negative bacteria and LAB dominated in the ‘aged’ retail cheeses before inoculation with *L. monocytogenes* strain No. 10 (day 0). During storage of the inoculated ‘aged’ cheeses for additional 20 days, spoilage LAB continued growth in all VP samples, while the growth of Gram-negative bacteria ceased or declined (Table 3). Levels of enterococci, total staphylococci and yeasts fluctuated greatly in the ‘aged’ batches on day 15 of retail storage. However, their counts never exceeded 7, 6 and 5 log_10_ CFU/g, respectively, after 20 days of additional storage (Table 3); thus, they were subdominant spoilers in the ‘aged’ whey cheeses. The mean pH of the ‘aged’ cheese batches declined further (*p* < 0.05) to a final pH range of 4.6 to 5.0 after 20 days of additional storage at 4 °C (Table 3). Thus, as it happened in most fresh batches before (Figure 1B), the natural whey cheese acidification by native LAB continued up to day 35 of total storage (Table 3). This progressive LAB acidification resulted in complete growth suppression of *L. monocytogenes* until the sell-by date of the ‘aged’ VP Anthotyros products of the second dairy plant (Table 3).

In an attempt to associate the great variations in pH and the growth pattern of *L. monocytogenes* with potential differences in the specific spoilage LAB and Gram-negative bacteria species prevailing in the Anthotyros products of the second dairy, an additional 44 representative LAB (Table 4) and 18 Gram-negative bacterial (Table 5) isolates, recovered from TSAYE, MRS and CFC plates of the fresh spoiling batches C and D on day 15 and the ‘aged’ batch X-15A, were identified according to Sameli et al. [37]. The majority of the prevalent LAB isolates (38/44; 86.4%) on day 15 were non-slime-producing *Leuconostoc* spp.; most of them (27/44; 61.4%) were assigned to three atypical *L. mesenteroides* biotypes L1–L3 (Table 4), also found to prevail in the terminal spoilage LAB association of batches C and D on day 40 (Table 2). However, while biotype L1 of *L. mesenteroides* predominated in batch D on day 15 (Table 4) and continued so on day 40 (Table 2), no biotype L1 isolates and only one biotype L2 isolate of *L. mesenteroides* were recovered from batch C on day 15. Most LAB isolates from batch C on day 15 were assigned to an additional *Leuconostoc* biotype L7, which fermented lactose, galactose, trehalose and raffinose but did not ferment L-arabinose and D-xylose (Table 4). Thus, L7 was an intermediate *Leuconostoc argentinum/lactis* biotype [41,42]. Notably, biotype L7 isolates were also recovered from the ‘aged’ batch X-15A, which mainly contained biotype L2 and L3 *L. mesenteroides* isolates (Table 4). Consistent with the results in Table 2, nonaciduric *C. maltaromaticum* was isolated from batch C, but not from batch D, on day 15; *C. maltaromaticum* was isolated from batch X-15A as well (Table 4). Finally, it should be stressed that no MRS or TSAYE isolates of *S. thermophilus* were recovered from batch D on day 15 (Table 4). However, all fresh (day 0) samples of batch D were confirmed to harbor an unexpectedly high (ca. 7-log_10_ CFU/g) population of a typical *S. thermophilus* starter biotype selectively grown/enumerated on MPCA and M17 agar plates at 37 °C (data not shown). Apparently, the fresh curd of batch D was somehow ‘postprocess contaminated’ heavily with those starter *S. thermophilus* cells, which might have multiplied to the 7-log level while the curd was still warm in the plastic molds transferred for draining.

Representative isolates of *Hafnia alvei* biotype I were exclusively isolated from batch C on day 15 (Table 5) and eventually became the predominant spoilage Gram-negative bacterium in batch C (Table 2) on days 30 and 40 of storage [37]. Conversely, *Klebsiella oxytoca* and, most surprisingly, *Shewanella putrefaciens* were recovered from batch D on day 15 (Table 5). Notably, *S. putrefaciens*, a specific spoilage species of animal foods at pH > 6.0, was not detectable in batch D by late storage (Table 2), probably because it was inhibited by the acid pH; instead, the spoilage association of batch D on days 30 and 40 was dominated by *H. alvei* and *Aeromonas* strains (Table 2). *H. alvei* and *Pantoea* sp. were recovered from the ‘aged’ spoiling samples of batch X-15A as well (Table 5).

## 4. Discussion

The significant growth of *L. monocytogenes* strain No. 10 in the fresh untreated (CN) Anthotyros samples during VP storage at 4 °C was consistent with previous studies indicating a high growth potential of this psychrotrophic pathogen in cold-stored whey cheeses [12,13,14,15,16,17,23] and confirmed the utmost need for in-package antimicrobial hurdles to control growth of postprocess listerial contamination in RTE Greek whey cheeses [15]. The mixed CFS of three Ent+ *E. faecium* strains (CEntE) evaluated as an alternative biopreservative to Nisaplin in this study had no immediate lethal antilisterial effects, because, apparently, it contained fewer active enterocin A-B-P molecules than the nisin molecules contained in the concentrated Nisaplin powder [15]. However, the antilisterial effectiveness of the CEntE appeared to increase after day 15, while the pH was declining below 5.5 during storage. The enterocin A-B-P activity against the target *L. monocytogenes* cells was probably enhanced at the low pH range of the naturally acidified Anthotyros cheeses, similarly to nisin and most enterocins in vitro and in situ in fresh acid-curd cheeses or other fermented cheese types [2,27,36,43]. In particular, the in situ antilisterial effects of *E. faecium* KE82 used as a bioprotective adjunct culture in fermenting cheese milks were stronger in the presence of a commercial starter culture containing *S. thermophilus* as the primary milk acidifier than in the respective cheese milks without addition of the starter [44]. In general agreement with the above studies, in this study, the antilisterial effects of the CEntE in most Ent-treated whey cheese samples increased with delay, which means that afterward, the native LAB biota, dominated by atypical *L. mesenteroides* strains, reduced the cheese pH below 5.5 to 4.5 to enhance the combined enterocin A-B-P activity. It is well known that the in situ effectiveness of the enterocins A and B increases when they are copresent to act synergistically [45,46].

Previous studies on the use of purified, semipurified or crude enterocin preparations for inhibiting *L. monocytogenes* in fresh, high-pH whey cheeses are scarce, if any; i.e., refer to the recent reviews by Silva et al. [27] and Falardeau et al. [2]. Most previous studies on this topic have dealt with soft acid-curd cheeses and other real or model cheese fermentations. Under these conditions, the native LAB present in the raw milk, or the starter LAB added to the milk after pasteurization, grow rapidly and acidify the milk, clotted with or without rennet. Rapid acidification creates an unfavorable low-pH niche for the growth of *L. monocytogenes*, whose inactivation is enhanced by adding enterocins or the viable Ent+ *Enterococcus* strains [40,47,48,49,50]. Conversely, in fresh whey cheeses stored under refrigeration, natural acidification is slow because of the lack of starter cultures and the reduced growth rates of the native LAB at cold temperatures. Therefore, a strong enterocin activity sufficient to cause major inactivation of *L. monocytogenes* in situ is very unlikely at least during early storage days when the natural whey cheese acidification is still mild. To our knowledge, there is only one previous challenge study by Aspri et al. [17] that found a complete 3 log_10_ CFU/g net inactivation of the inoculated target strain *L. monocytogenes* 33,413 in fresh Cyprian Anari whey cheese, after coinoculation with 6 log_10_ CFU/g of the EntA/B+ *E. faecium* DM 33 and aerobic storage of the cheeses at 4 °C for 9 days only. Interestingly, *E. faecium* DM 33 strain, which was listericidal, and two additional EntA/B+ *E. faecium* strains, which exerted listeriostatic effects in Anari under the same challenge conditions, originated from raw donkey milk [17].

In this study, neither was the growth of inoculated *L. monocytogenes* strain No. 10 prevented nor was the early growth of psychrotrophic spoilage *Enterobacteriaceae* and *Pseudomonadaceae* suppressed. However, the increasing growth competition and prevalence of naturally selected members of *Lc. mesenteroides, Lc. lactis* and *C. maltaromaticum* exerted a mitigation effect against the pathogen and the Gram-negative spoilage community in the ‘aging’ Anthotyros whey cheeses after day 15 of VP/4 °C storage. In particular, the populations of *L. monocytogenes* never increased above 6.5 log units, not even in the untreated (CN) samples of batches A to D. Significant growth inhibition or mitigation effects against Gram-negative spoilage bacteria have been attributed to the use of *Carnobacterium* spp. in the form of single selected adjunct strains or commercial bioprotective cultures (i.e., Lyofast CNBAL, Lyofast FPR2) in fresh MAP Ricotta cheese [25,26] and another Italian fresh cheese type [51]. Lyofast CNBAL containing *Carnobacterium* spp. was the best performing commercial culture against spoilage *Pseudomonas* spp. in fresh Ricotta cheese [26]. However, the primary role of the Lyofast CNBAL bioprotective culture marketed by Sacco System (Italy) is a strong (approximately 4-log_10_ CFU/g) inhibition of *L. monocytogenes* growth by its Ent+ *Carnobacterium* spp. strain constituents in situ in cheese ripened at low temperatures and without sugar [52]. In this study, there was no evidence that the presence of *C. maltaromaticum* in batches B and C was associated with bacteriocin (carnocin)-mediated antilisterial effects in situ. However, native Bac+ strains within the most prevalent *Lc. mesenteroides, Lc. lactis* and *C. maltraromaticum* isolates might exist to have contributed to the retardation or the strongest inhibition of *L. monocytogenes* growth in the CN samples of batches B and D, respectively (Figure 1A). Although the production of bacteriocins, such as mesentericin Y105, by dairy *Leuconostoc* spp. has been well established [53], applications of the producer strains or of *Leuconostoc* bacteriocin preparations to cheese and other dairy products have so far been very limited [2,27,43]. Conversely, few promising applications of bacteriocins or Bac+ strains of *S. thermophilus* and *S. macedonicus* against cheese spoilage bacteria have been reported [27,43], including a recent study on the antibacterial effects of in situ produced thermophilin T by *S. thermophilus* ACA-DC 0040 in fermented whey used for Myzithra cheese production [54]. This particular application is emphasized because, in this study, the increased growth inhibition of *L. monocytogenes* in batch D during storage (Figure 1A) might have been due to beneficial metabolic activities of the excessive *S. thermophilus* ‘natural contamination’ that had occurred during the handling of the fresh batch D curd in the plant for 24 h before VP storage under refrigeration [37]. Additional studies are required to elucidate the potential presence and impact of Bac+ isolates in fresh Anthotyros cheeses.

The native microbial community as a whole (i.e., termed microbial consortia that include adventitious LAB and various other Gram-positive and Gram-negative bacterial species and yeasts) has been reported to be more effective against *L. monocytogenes* than individual strains alone or artificial simplified consortia of few selected strains [2,55,56,57]. However, the microbial consortia mechanisms and specific strain interactions behind the in situ inactivation or growth inhibition of *L. monocytogenes* in fresh cheeses are still unclear [2]. In this study, apart from the prevalent native LAB biota dominated by several diverse *Lc. mesenteroides* biotypes followed by *C. maltraromaticum* and *Lc. lactis/argentinum* biotypes, the Gram-negative biota appeared to be significant not only as spoilers but also as modulators of the growth/survival responses of *L. monocytogenes* in the fresh ‘aging’ Anthotyros cheese batches in situ. Indeed, based on their isolation frequencies, lowered levels of Gram-negative bacteria with a higher prevalence of nonfermentative aerobic *Pseudomonas* and related species (i.e., prevalent in batch D and selected by the CEntE in batches A and B; Table 2) were associated with greater declines of *L. monocytogenes* (Figure 1A). The above positive interactive effects probably developed because the aerobic *Pseudomonas*-like spoilage bacteria, unable to ferment lactose, might have exerted a reduced nutrient (sugar) competition against the LAB biota growing primarily at the expense of fermentable lactose. In contrast, high levels of psychrotrophic fermentative enterobacteria, mainly *H. alvei* and *S. liquefaciens*, likely enhanced growth and survival of *L. monocytogenes* in batch C and in the CN samples of batches A and B. *H. alvei* and *S. liquefaciens* probably managed a fast lactose uptake and thus exerted increased competitive ‘early’ growth effects against LAB at 4 °C under vacuum. Overall, *Pseudomonas putida,* other *Pseudomonas* spp., *S. liquefaciens* and mainly *Hafnia-Obesumbacterium* have been genotyped within the most prevalent members of the very diverse non-LAB communities in various fresh cheese products [51,56,57,58]. Moreover, the above primary Gram-negative genera along with *Shewanella* spp. and *Flavibacterium* spp. were members of the spoilage microbial community of the whey used in Ricotta cheese production [21]. Particularly *H. alvei* was shown to have a high ecological and aromatic impact as part of the whole microbial community in an experimental smear soft cheese [56] and in pasteurized Ricotta cheese whey batches [21]. *H. alvei* appears to have a similar high impact in traditional Greek whey cheeses because it has been the most prevalent Gram-negative bacterium in the microbial spoilage community of Anthotyros [37] and Manouri [59]. Additional in-depth studies at the strain level are required to elucidate growth interactions between the most prevalent spoilage LAB and Gram-negative bacteria species or biotypes in fresh Ent-treated and CN Anthotyros cheese batches and their specific effects on the behavior of *L. monocytogenes* during VP storage at 4 °C or higher abusive storage temperatures.

In summary, postprocessing contamination of Anthotyros and other Greek whey cheeses with *L. monocytogenes* remains a food safety concern, although the percentage of contaminated market cheese products appears to be decreasing during the last three decades due to improvements in the equipment and the cheese processing and storage conditions [28,29]. Notably, 15.4% of the retail whey cheese samples collected from the Greek market in 2010 had pH < 4.4 and, thus, would not permit the growth of *L. monocytogenes* [29]. Angelidis et al. [29] stated that such low pH values for whey cheese products are unexpected based on their manufacturing technology and, therefore, are associated with lower quality and reduced shelf life. In our opinion, altogether, the results of the present study reveal that such low pH values indicate that those commercial whey cheese samples were already ‘aged’ for two or more weeks in the supermarket freezers before their sampling, and some of them were likely to have been temperature abused during retail distribution and storage. Abusive or extended cold storage periods of whey cheeses not only result in short residual shelf life, but also may provide misleading results as regards the behavior of *L. monocytogenes* in case they are used as RTE market samples in challenge tests. For instance, a recent comprehensive study assessing the capacity of growth of *L. monocytogenes* in various cheeses from the Greek market included the whey cheeses Anthotyros and Manouri among those that did not support the growth of *L. monocytogenes* due to the high microbial competition and low pH during vacuum storage at 7 °C, thus mimicking temperature abuse [9]. In specific, the mean pH values of those Anthotyros cheeses recorded at the beginning (day 0), middle (day 8) and end (day 16) of storage were 6.72, 5.27 and 4.84, respectively [9]. Thus, the latter two pH values were very acidic, contrary to the corresponding mean pH values of the present fresh VP Anthotyros cheese samples without enterocin that were much higher, i.e., 6.80 (day 0), 6.84 (day 8) and 6.21 (day 15), during storage at 4 °C (Table 1). However, in that previous challenge study of Kapetanakou et al. [9], the native LAB and other bacteria, claimed by the authors as responsible for the natural whey cheese acidification and overall microbial competition, were not identified. Nonetheless, this study clearly demonstrated that none of the ‘aged’ (15-day-old) retail batches permitted *L. monocytogenes* growth, due to the reduction in their pH by an already high population level of native psychrotrophic LAB biota at the beginning of the additional 20-day VP storage period at 4 °C (Table 3), mainly consisting of *Lc. mesenteroides*, *Lc. lactis* and *C. maltaromaticum* (Table 4). However, challenge testing conditions of an ‘aged’ whey cheese rather represent an unrealistic scenario because the highest possibility for accidental natural contamination of a fresh RTE whey cheese with *L. monocytogenes*, as well as with all types of spoilage bacteria interacting with the pathogen’s growth, exists before or during vacuum packaging or MAP in individual plastic bags or trays in the plant prior to retail distribution [18,21,60,61]. Of course, an ‘aged’ whey cheese may be cross-contaminated with the pathogen after opening at home and storage in a domestic refrigerator, but this possibility and the safety risks associated with this possibility seem negligible. Nevertheless, care is needed when selecting fresh (whey) commercial cheese samples for challenge tests. In addition, their industrial or traditional manufacturers should be interviewed as regards the potential use of any chemical or natural preservatives or commercial bioprotective cultures. In this challenge study, we did ensure that the isolated *C. maltaromaticum* strains were not of commercial culture origin and no other types of biopreservatives had been intentionally added in the fresh Anthotyros whey cheeses studied.

## 5. Conclusions

In conclusion, this challenge study validated an average 2-log growth potential of *L. monocytogenes* in Greek Anthotyros whey cheeses when the artificial contamination with the pathogen was introduced in 50 g portions of fresh (1-day-old) retail cheese batches (mean pH 6.8; mean LAB level < 3.0 log_10_ CFU/g) before vacuum packaging and storage at 4 °C for up to 40 days. In contrast, the pathogen failed to promote major growth (<0.5 log increase) when it was artificially contaminated in Anthotyros batches that had been ‘aged’ (mean pH 6.1; mean LAB level >7.5 log_10_ CFU/g) in retail for 15 days after manufacture and then stored as above for an additional 20 days at 4 °C. Overall, the results revealed the vital importance of an unmonitored whey cheese acidification by native LAB, predominantly *Lc. mesenteroides*, in controlling growth of *L. monocytogenes* after a preceding storage at 4 °C in VP for approximately two weeks. In specific, listerial growth was suppressed after the whey cheese acidification reached a pH of 5.5, while listerial survival declined at pH 5.0 to 4.5 in most naturally acidified cheese batches after 30 to 40 days of storage. The addition of 5% CEntE in the fresh cheeses reduced the survival of *L. monocytogenes* by late storage, probably because the antilisterial activity of the enterocin A-B-P mix was enhanced at low pH in situ. However, neither the CEntE nor the native LAB biota could prevent a major (>1–2 log_10_ CFU/g) growth of *L. monocytogenes* during the first (mainly) and second weeks of VP storage at 4 °C. Regardless of CEntE addition, the growth/survival of *L. monocytogenes* in fresh Anthotyros was strongly batch-dependent and likely plant-dependent. Overall, higher postprocess contamination levels of native LAB, accompanied by cogrowth of a non-LAB spoilage microbiota dominated by *Pseudomonas* spp. or related nonfermentative Gram-negative bacteria, was associated with earlier growth cessation times and lowered surviving populations of *L. monocytogenes*, whereas high early growth levels (>8.0–8.5 log_10_ CFU/g) of spoilage fermentative psychrotrophic *Enterobacteriaceae*, predominantly *Hafnia alvei*, were likely associated with increased and more extended growth of *L. monocytogenes* during storage. Further research is required to assess potentially increased antilisterial effects during refrigerated storage of traditional Greek fresh whey cheeses by the application of a more active enterocin A-B-P concentrate than the CEntE (400 AU/mL) herein, singly or in combination with bioprotective (Bac+) strains derived from the native psychrotrophic whey cheese spoilage LAB biota.

## Figures and Tables

**Figure 1 foods-11-00334-f001:**
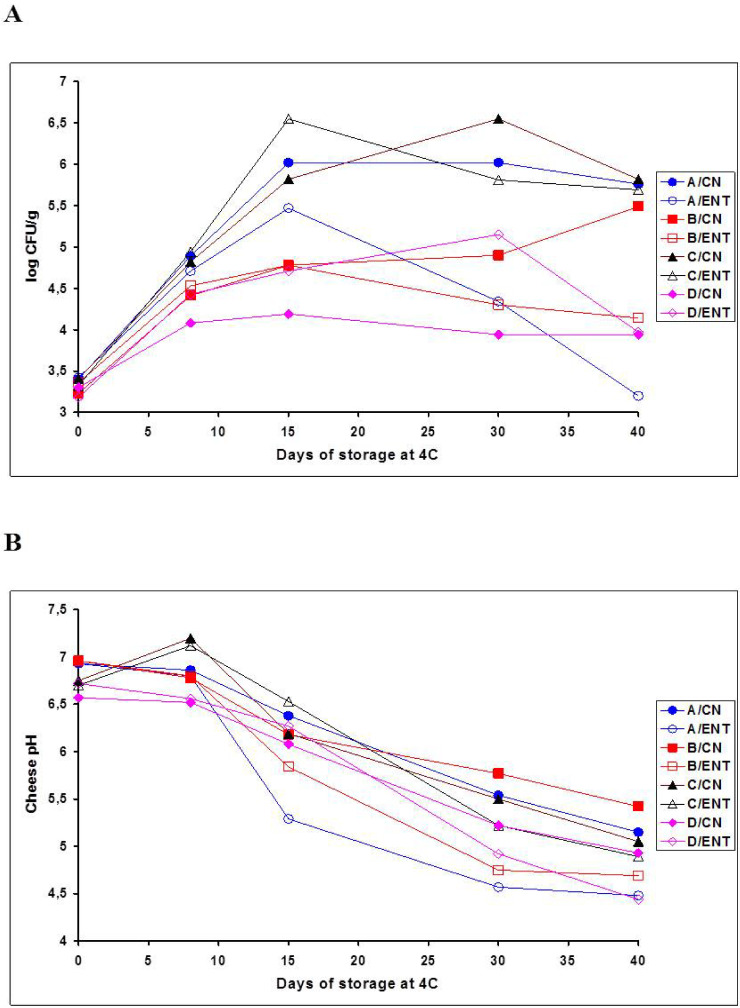
Cheese batch-dependent growth and survival of *Listeria monocytogenes* (**A**) in association with the pH changes (**B**) during refrigerated (4 °C), vacuum-packed storage of four batches (A, B, C and D) of fresh Greek Anthotyros whey cheeses without (CN) or with 5% *(v/w*) of an added crude enterocin A-B-P extract (ENT). The pH data in Figure 1B were adapted from Table S1 [37].

**Figure 2 foods-11-00334-f002:**
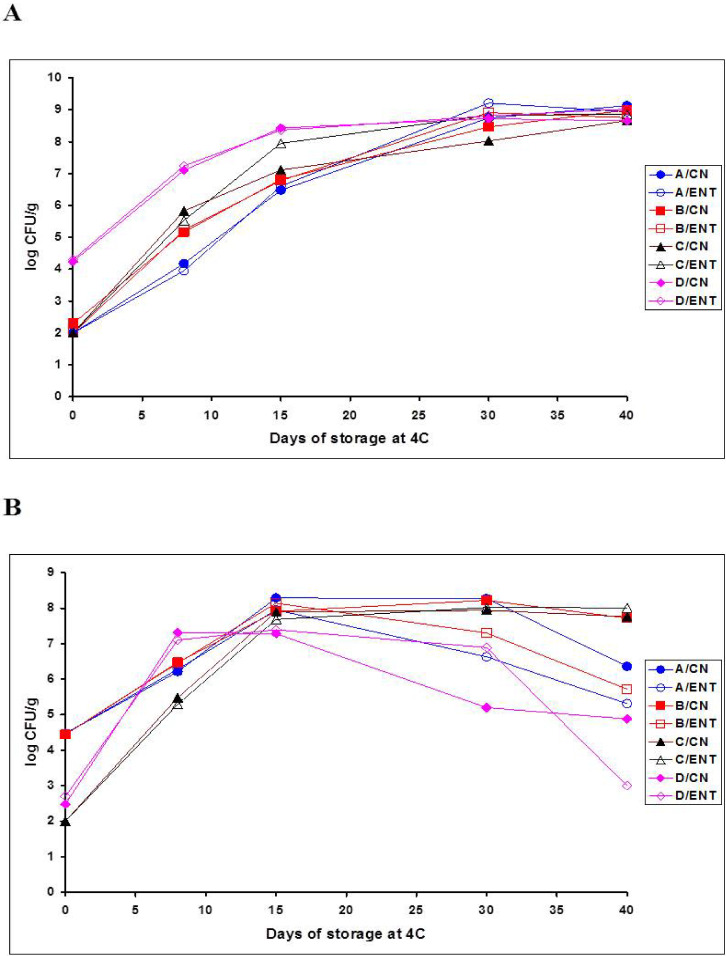
Cheese batch-dependent growth (log_10_ CFU/g) of spoilage lactic acid bacteria, enumerated on MRS agar at 30 °C (**A**), and spoilage *Pseudomonas*-like bacteria, enumerated on CFC agar at 25 °C (**B**), during refrigerated (4 °C), vacuum-packed storage of four batches (A, B, C and D) of fresh Greek Anthotyros whey cheeses without (CN) or with 5% *(v/w*) of an added crude enterocin A-B-P extract (ENT).

**Table 1 foods-11-00334-t001:** Behavior of the inoculated *Listeria monocytogenes* (log_10_ CFU/g) populations in association with the evolution of the primary spoilage bacterial groups and the pH changes during refrigerated (4 °C) storage of fresh, vacuum packaged Anthotyros whey cheeses without (CN) or with (CEntE) addition of 0.5% crude enterocin A-B-P extract ^a^.

Bacterial Group	Cheese Treatment	Days of Storage at 4 °C				
		0	8	15	30	40
*Listeria monocytogenes* strain No. 10	CN	3.34 a	4.55 b	5.20 b	5.35 b	5.25 b B
		(0.09)	(0.38)	(0.87)	(1.17)	(0.89)
	CEntE	3.32 a	4.65 b	5.38 c	4.90 cb	4.25 ab A
		(0.10)	(0.22)	(0.85)	(0.72)	(1.04)
Total dairy lactic acid bacteria (LAB)—selective colony enumeration on MPCA plates at 37 °C for 48 h	CN	4.28 a	6.70 ab	7.17 b AB	8.39 c B	8.67 c B
		(2.06)	(1.12)	(0.97)	(0.27)	(0.52)
	CEntE	4.00 a	6.12 a	8.32 b B	8.68 b B	8.66 b B
		(2.13)	(1.21)	(0.63)	(0.61)	(0.29)
Total Gram-negative dairy bacteria—selective colony enumeration on MPCA plates at 37 °C for 48 h	CN	4.51 a	6.24 b	6.67 bc A	7.46 c A	7.05 bc A
		(1.15)	(1.03)	(1.61)	(0.69)	(0.45)
	CEntE	4.97 a	5.85 b	7.14 c AB	7.52 c A	6.89 bc A
		(1.29)	(1.25)	(1.31)	(0.60)	(0.89)
**Whey cheese pH ^b^**	CN	6.80 d	6.84 d	6.21 c	5.51 b B	5.14 a B
		(0.18)	(0.28)	(0.13)	(0.23)	(0.21)
	CEntE	6.83 c	6.82 c	5.98 b	4.87 a A	4.63 a A
		(0.14)	(0.23)	(0.54)	(0.28)	(0.21)

^a^ Values are the means of four independent whey cheese batches (*n* = 4); standard deviation values are shown in brackets. Within a row, means lacking a common lowercase letter are significantly different (*p* < 0.05). Within a column for each analysis, means bearing an uppercase A versus B are significantly different (*p* < 0.05); ^b^ Whey cheese pH data are adapted from Sameli et al. [37].

**Table 2 foods-11-00334-t002:** Overview of the terminal spoilage association in four batches (A, B, C and D) of fresh Anthotyros whey cheese without (CN) or with 5% crude enterocin A-B-P extract (Ent) after 30 to 40 days of vacuum-packaged storage at 4 °C (data adapted from Sameli et al. [37] were suitably modified to present the numerical distribution of the isolates per CN or Ent treatment within each whey cheese batch).

Species	Biotypes	Whey Cheese Batch Isolates								Total Isolates (%)
		A		B		C		D		
		CN	Ent	CN	Ent	CN	Ent	CN	Ent	
*Leuconostoc mesenteroides* (Non-slime-producers)	3	10	8	11	10	7	9	10	10	75 (62.5)
*Leuconostoc mesenteroides* (Slime-producers)	1	4	7	2	2	2				17 (14.2)
*Leuconostoc lactis*	1						2		2	4 (3.3)
*Carnobacterium maltaromaticum*	4			2	3	4	4			13 (10.9)
*Streptococcus thermophilus*	1							5	2	7 (5.9)
*Enterococcus faecium*	1								1	1 (0.8)
*Lactococcus lactis*	1					2				2 (1.6)
Mesophilic *Lactobacillus* sp.	1	1								1 (0.8)
Total LAB isolates		15	15	15	15	15	15	15	15	120

*Hafnia alvei*	2					13	17	2	9	41 (42.7)
*Serratia liquefaciens*	2	3		7	3	3	1			17 (17.7)
*Rahnella aquatilis*	2	5		3						8 (8.3)
*Pantoea* sp.	1					1				1 (1.1)
*Klebsiella oxytoca*	2					1		2		3 (3.1)
*Enterobacter* sp./*E. cloacae*	2					1	1	1	1	4 (4.2)
*Pseudomonas* sp.	3	2	5		4					11 (11.4)
*Aeromonas* sp.	2							9		9 (9.4)
*Flavibacterium* sp.	1							2		2 (2.1)
Total Gram-negative bacteria isolates		10	5	10	7	19	19	16	10	96

**Table 3 foods-11-00334-t003:** Behavior of inoculated *L. monocytogenes* (log_10_ CFU/g) in association with the growth of the spoilage microbiota (log_10_ CFU/g) and the pH changes during additional refrigerated (4 °C) VP storage of the ‘aged’ (15-day-old) retail Anthotyros whey cheeses ^a^.

Bacterial Group	Enumeration Medium/Incubation Conditions	Days of Additional Storage at 4 °C		
		0 (15)	8 (23)	20 (35)
*Listeria monocytogenes* No. 10	PALCAM agar/30 °C; 48–72 h; aerobically	3.13 a (0.28)	3.29 a (0.31)	3.01 a (0.23)
Total spoilage lactic acid bacteria (LAB)	M17 agar/22 °C; 72 h; aerobically	8.29 a (0.25)	8.64 b* (0.07)	8.49 ab* (0.56)
Total spoilage Gram-negative bacteria		7.70 a (0.10)	7.29 a* (0.96)	7.18 a* (1.03)
Total mesophilic aciduric LAB	MRS agar/30 °C; 72 h; aerobically	7.74 a (0.89)	8.09 a (0.72)	7.99 a (0.79)
*Pseudomonas* and related Gram-negative bacteria	Cephalothin–fucidin–cetrimide (CFC) agar/ 25 °C; 48 h; aerobically	7.60 b (0.25)	6.48 ab (1.89)	6.13 a (1.99)
Coliform bacteria	Violet Red Bile agar (VRBA)/37 °C; 18–24 h; anaerobically (double-layered)	6.70 b (0.92)	5.95 ab (1.91)	5.00 a (1.20)
Enterococci	Kanamycin Aesclulin Azide (KAA) agar/ 37 °C; 48 h; aerobically	4.11 a (2.12)	4.63 a (1.70)	4.67 a (1.74)
Total staphylococci	Baird-Parker agar with egg yolk tellurite/ 37 °C; 48 h; aerobically	4.65 a (1.36)	4.62 a (1.37)	3.43 a (1.29)
Yeasts	Rose Begnal Chloramphenicol (RBC) agar/ 25 °C; 48 h; aerobically	2.58 a (0.53)	2.65 a (0.16)	3.69 a (1.21)
**Whey cheese pH**		6.10 b (0.16)	5.14 a (0.35)	4.86 a (0.24)

^a^ Values are the means of three independent cheese batches *(n* = 3); standard deviation values are shown in brackets. Within a row, means lacking a common lowercase letter are significantly different (*p* < 0.05). Within a column for each analysis, means bearing an asterisk are significantly different (*p* < 0.05).

**Table 4 foods-11-00334-t004:** Biochemical characterization of additional 44 representative LAB isolates recovered from MRS and TSAYE agar plates of spoiling (day 15) Anthotyros whey cheese products stored at 4 °C in vacuum.

Biochemical Test	*Leuconostoc* spp. Isolates					Other LAB Isolates			Total Isolates
	L1	L2	L3	L6	L7	Cn	Lb	Ent	
Cell morphology	BC	BC	BC	BC	BC	SR	C	C	
CO_2_ from glucose	+	+	+	+	+	−	−	−	
NH_3_ from arginine	−	−	−	−	−	(+)	−	++	
Growth at 45 °C	−	−	−	−	−	−	−	+	
Growth in 4% NaCl	+	+	+	+	+	+	+	+	
Growth in 6.5% NaCl	+	+	+	+	+	−	+	+	
Slime from sucrose	−	−	−	−	−	−	−	−	
Growth on KAA agar	−	(+)	−/(+)	−	−	+	−/(+)	++	
Acid from:									
Maltose	6/12	+	1/6	−	+	+	+	+	
Mannitol	−	−	−	−	−	+	+	+	
Lactose	+	+	+	+	+	+	+	+	
Ribose	−	(+)/+	1/6	−	−	+	−	+	
L-Arabinose	−	+	+	−	−	−	+	+	
Xylose	+	+	+	+	−	−	+	−	
Raffinose	−	+	−	+	−/+d	−	+	−	
Sucrose	+	+	+	+	+	+	+	+	
Cellobiose	−	4/9	1/6	−	5/10	+	+	+	
Trehalose	+	+	+	+	+	2/3	+	+	
Galactose	+	+	+	+	+	+	+	+	
Total isolates	12	9	6	1	10	3	2	1	44
Batch C (CN/Ent)	0/0	1/0	0/0	0/1	3/4	1/0	1/1	0/0	12
Batch D (CN/Ent)	6/5	0/0	0/0	0/0	0/0	0/0	0/0	0/1	12
Batch X-15A	1	8	6	0	3	2	0	0	20

+, positive reaction; −, negative reaction; (+), weak positive reaction; +d, delayed positive reaction; ++, strong positive reaction; 6/12, 6 out of 12 isolates were positive. Biochemical characterization: L1, L2, L3 and L6, *Leuconostoc mesenteroides* (different dairy biotypes); L7, *Leuconostoc argentinum/Leuc. lactis*; Cn, *Carnobacterium maltaromaticum*; Lb, unidentified mesophilic *Lactobacillus* sp.; Ent, *Enterococcus faecium.*

**Table 5 foods-11-00334-t005:** Identification of additional 18 representative isolates of Gram-negative bacteria recovered from CFC and TSAYE agar plates of spoiling (day 15) VP Anthotyros whey cheese products at 4 °C, as determined by the API 20E identification method.

Test	Reactions/Enzymes	*Hafnia* *alvei* I	*Pantoea* sp.	*Klebsiella* *oxytoca*	*Shewanella* *putrefaciens*	Total Isolates
ONPG	β-Galactosidase	+/−	+	+	−	
ADH	Arginine dihydrolase	−	−	−	−	
LDC	Lysine decarboxylase	+	−	+	−	
ODC	Ornithine decarboxylase	+	−	−	−	
CIT	Citrate utilization	+	−	+	−	
H_2_S	H_2_S production	−	−	−	+	
URE	Urease	−	−	−	−	
TDA	Tryptophane deaminase	−	−	−	−	
IND	Indole production	−	−	+	−	
VP	Acetoin production	+	−	+	−	
GEL	Gelatinase	−	+	−	+	
GLU	Glucose (F/O)	+	+	+	−	
MAN	Mannitol (F/O)	+	+	+	−	
INO	Inositol (F/O)	−	−	+	−	
SOR	Sorbitol(F/O)	−	−	+	−	
RHA	Rhamnose (F/O)	+	−	+	−	
SAC	Saccharose (F/O)	−	+	+	−	
MEL	Melibiose (F/O)	−	−	+	−	
AMY	Amygdalin (F/O)	−	−	+	−	
ARA	Arabinose (F/O)	+	+	+	−	
OX	Oxidase reaction	−	−	−	+	
	API code	4305112 5305112	1006122	5245773	0402004	
	Identification accuracy	Excellent Very good	Acceptable	Good	Excellent	
	Total isolates	8	4	1	5	18
	Isolates/batch C (CN/Ent)	3/3	0/0	0/0	0/0	6
	Isolates/batch D (CN/Ent)	0/0	0/0	0/1	3/2	6
	Isolates/batch X-15M	2	4	0	0	6

+, positive colored reaction; −, negative colored reaction, according to the API 20E manual instructions.

## Data Availability

The data presented in this study, as well as the bacterial isolates identified, are available on request from the corresponding author.

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
