# Peer review of "Growth and Biocontrol of Listeria monocytogenes in Greek Anthotyros Whey Cheese without or with a Crude Enterocin A-B-P Extract: Interactive Effects of the Native Spoilage Microbiota during Vacuum-Packed Storage at 4 °C"

_foods, 2022, doi:10.3390/foods11030334_

Round 1

Reviewer 1 Report

Dear Authors

The article investigated the behavior of L. monocytogenes in Anthotyros whey cheese in the presence or absence of crude enterocin A-B-P extracted. 

The topic is interesting. However, the manuscript must be reviewed by a native speaker. Moreover, while the results of some experiments are mentioned in the "Results" section, the method of related analysis is not stated in the "Materials and Methods" section (such as Question 1). 

Q1. How was the method for assessment of enterocin titer? Where are the related results?

Q2. Please explain or give a flow chart for the manufacturing process of Anthotyros cheese.

Q3. What are the standard deviations in Fig 1 A & B.

Q4. How did you ensure the homogenous distribution of enterocin in samples?

Q5. What is the effect of vaccum packaging on the growth and survival of the studied microorganism?

Sincerely, 

Author Response

Dear Reviewer 1

Thanks for your comments. Please see our responses to the attached file.

John Samelis

Reviewer 2 Report

This manuscript provides novel information on the effect of enterocins and naturally occurring contaminating microbiota on the survival of Listeria monocytogenes in Greek Whey Cheese. The study is well designed. The manuscript is well written but some of the wording could be improved and should be reviewed by a native English speaker.  The introduction is well-structured and explains the reason for the study; however, it needs to be shortened.  The discussion can lose the interest of the reader and needs to be shortened.

Specific comments

Section 2.1 last paragraph - “Relative to this study, the CFS of all three strains after culturing in MRS or M17 broth at 30 °C possessed a strong in vitro antagonistic, enterocin (Ent+) activity against several L. monocytogenes indicator strains, including strain Scott A [39].  Enterocin activity was retained after CFS neutralization, and ranged from strongly listeriostatic to listericidal depending on the pH and the culture conditions prevailing in co-growth challenge experiments in the above two synthetic media and skim milk [39].” This belongs in a discussion or introduction, or in the results section if it was work that was done in this study and should not be in the methods.

Section 2.3 first paragraph – remove the hyphen between numbers and units, ie “500-g shoft cheese” should be “500 g soft cheese”, and the same with “10-mL culture”.  Please check entire manuscript.

Section 2.3 first paragraph “artificially contaminated with L. monocytogenes no.10 by”.  Please rephrase to L. monocytogenes strain no.10.  It took a bit of effort to figure out what no.10 meant.

Section 2.3 first paragraph “diluted to yield ca. 3 logs of listerial”.  Change the abbreviation “ca.” to a word, such as “approximately”.  “3 logs” is too informal.  Change to “3 log10”. Please review the manuscript for further occurrences of this, such as the paragraph on p 8.

Section 2.4 Neogen Culture Media is described as formerly being LAB M. In the previous sections, reference is made to Lab M.  Was the media in those sections purchased when the supplier was still Lab M or had it changed to Neogen?  Please review previous sections and update to Neogen if applicable.

Section 2.6 converted to log10 CFU/g

Section 3.1 and elsewhere.  Please add the subscript 10 to log and indicate unit, ie log10 CFU/g.

Table 1 Please change the column 1 heading.  “Whey cheese pH” is not a Bacterial Group. Also, the growth conditions for LAB and Gram-negatives do not need to be in the row descriptions.  This uses more space and is out of place considering the equivalent information is not provided for L. monocytogenes.

Figure 1.  The “A” and “B” referring to L. monocytogenes and pH, respectively, is confusing given that the batches are A, B, C and D.  Perhaps label the graphs using lower case letters, ie a and b.

Table 2. Please indicate that A, B, C and D are the batches, ie “Overview of the terminal spoilage association in four batches (A, B, C and D) of”.

Table 3. Whey cheese pH is not a Bacterial Group.

Discussion p 17 “In accordance, the lack of starter cultures, the high pH values and the reduced growth rates of native LAB in fresh whey cheeses stored under refrigeration are against to obtain a strong full enterocin activity against L. monocytogenes in situ, at least during early storage days when the natural whey cheese acidification is still mild.”. Please rephrase this sentence.  It does not make sense. 

Author Response

Dear Reviewer 2

Thanks for your comments. Please see our responses to the attached file.

John Samelis
